# The Shear Behavior of Insulated Precast Concrete Sandwich Panels Reinforced with BFRP

Xia Liu [ID], Xin Wang *[ID], Tan Yang and Zhishen Wu

Key Laboratory of C & PC Structures Ministry of Education, Southeast University, Nanjing 211189, China
* Correspondence: xinwang@seu.edu.cn

**Abstract:** Typical insulated precast concrete sandwich panel (PCSP) systems are composed of two concrete wythes separated by a layer of insulation. The structural behavior of insulated PCSP systems heavily depends on elements between two wythes known as connectors, which ensure they work as a whole. Double shear tests were carried out on 58 insulated PCSP specimens reinforced with basalt fiber-reinforced polymer (BFRP) connectors; failure modes, load displacement curves and bearing capacity of BFRP connectors were obtained. Effects of diameter, insulation thickness, installation angle, layout spacing and combined action on shear capacity were analyzed. The results show that the span ratio of the connector was suggested to be less than 15, and the angle of the connector should be set to 60° or 75° for suitable stiffness and bearing capacity with an abundant safety margin. The shear capacity decreases slightly with the increase in the connectors' spacing while the overall impact is small. The shear force of connectors in a plane or spatial combination can be calculated according to that of one single connector. Moreover, the shear capacity model of BFRP connectors proposed in this paper provides a favorable design option for insulated PCSP systems using BFRP connectors.

**Keywords:** BFRP connector; shear capacity; span ratio; installation angle; combined action; shear capacity model

## 1. Introduction

Typical insulated precast concrete sandwich panel (PCSP) systems are composed of two concrete wythes separated by a layer of insulation [1]. Connectors penetrate the insulation layer and join the two concrete wythes to ensure they work together [2]. The structural properties of PCSP systems depend largely on connectors, which are designed to carry gravity loads from floors or roofs and to resist lateral loads caused by wind as well as external effects caused by earthquake. Connectors are known to be divided into three types, including ordinary steel connectors, alloy connectors and fiber-reinforced polymer (FRP) connectors [3]. Though ordinary steel bar connectors have advantages of low cost and convenient installation, the thermal bridge effect could not be avoided due to their relatively high thermal conductivity [4]. Therefore, it is difficult to meet the requirements of energy-saving indicators for insulated PCSP systems [5]. In addition, the poor corrosion resistance of steel connectors was a potential safety hazard to insulated PCSP systems. Compared to steel connectors, the alloy connectors have good corrosion resistance, durability and low thermal conductivity, while the lifecycle costs of this type were significantly increased [6]. In the recent two decades, FRP connectors have leaped forward due to their high strength and stiffness, as well as proven durability combined with lower thermal conductivity relative to conventional metallic connectors [7]. Generally, FRP has a thermal conductivity approximately 1/150 that of steel and 1/30 that of concrete [8]. Therefore, it will have broad engineering application prospects in the field of ultra-low-energy residential engineering [9–11].

FRP connectors in insulated PCSP systems have gained widespread attention nowadays. Experimental study on fourteen different types of commercially available FRP ties

has been conducted [12]. Their failure modes and responses were quantified, and the simplified engineer-level multilinear strength curves were developed for each connection. Xue et al. [13] have proved the efficiency of FRP connectors by corrosion test in a concrete alkali environment with a temperature of 60 °C, where the safety factor of shear resistance could reach 8.0 after 183 aging days. Three types of GFRP connectors produced from available sand-coated and threaded rods were tested and compared to conventional steel as well as polymer connectors [14]. The shear strength of GFRP connectors ranged from 60 to 112 MPa, which reach 1/3 that of conventional steel connectors. Forty-six segments representing precast concrete sandwich panels using CFRP grid connectors were adopted by Hodicky et al. [15], and the shear transfer mechanism of the CFRP grid connectors was elaborated. Tomlinson et al. [16] studied BFRP-reinforced partially composite precast concrete insulated wall panels with BFRP shear connectors, indicating that the tested walls have between 60% and 90% of the strength of all-steel reinforcement and connectors. Thirty-eight push-through tests were performed on a precast concrete insulated sandwich panel using combined angled and horizontal BFRP connectors [17]. The research indicated that the steel and BFRP connectors have similar strengths in tension stress mode; both the strength and stiffness increase with the connector angle and diameter. In order to evaluate the bonding performance between BFRP connectors and concrete wythes, the pull-out test with surface winding and sandblasting connectors have been carried out, where excellent bonding performance was evidenced with the bonding stress of 13.9 MPa [18]. Salam Al-Rubaye et al. [19] studied four different types of GFRP connectors on their stiffness, strength and ductility, and the shear failure mechanism of different connectors has been revealed. The precast sandwich wall panel using recycled aggregate and angled BFRP shear connectors was developed by Xie et al. [20]. Notable advantages in terms of the bearing capacity and ductility over panels with commercial GFRP needle connectors have been evidenced. Choi et al. [21] used GFRP grids to enhance both the structural and thermal performances of insulated concrete sandwich wall panels. All specimens exhibited sufficient flexural performance satisfying the strength presented in the current code, demonstrating 2.70–3.63 times higher strength capacities than the design load. Diagonal shear connectors of GFRP bars with diameter of 4 mm were adopted by Sylaj et al. [22], while the governing failure mode for all panels is the fracture of the GFRP shear connectors.

In fact, the anchorage length of connectors in concrete is relatively small due to the thickness limit of concrete wythes. Thus, a series of structural measures have been taken on FRP connectors to improve the mechanical anchorage capacity, including increasing the spacing between the vertical lines of the CFRP grid [15], slotting on panel-type FRP connector [23], "Cross" design at both anchorage ends of connectors [12], crisscross for rod connector [19], surface winding or sandblasting [18], etc. However, CFRP grids were susceptible to tension rupturing and compression buckling with the increased spacing. While for panel-type connectors, the moments of inertia in two directions were different, which has a great impact on their mechanical performance in PCSP systems. Moreover, the FRP connectors with "Cross" design have the disadvantage of complex manufacture and installation, and it is easy to cause damage to the insulation layer due to their expanded tips. The surface winding or sandblasting treatments improved the bond performance while the cost increased. Therefore, an improved integrated deep-ribbed BFRP connector manufactured by pultrusion process was introduced in this paper. This type of BFRP connector has benefits of simple structure, convenient installation, low thermal conductivity and excellent bonding performance with the concrete, which will provide an alternative scheme for the selection of connectors. There is a primary need to explore influencing factors and to develop constitutive laws for the BFRP connector in insulated PCSP systems.

For this purpose, the paper presents an experimental investigation of segments representing a typical PCSP, as shown in Figure 1. A total of 58 specimens using BFRP connectors were adopted in double shear tests to obtain failure modes, load displacement curves and shear capacity. Parameters of the BFRP connector's diameter, thickness of insulation layer, installation angle, layout spacing and combined action were analyzed. Test results were

used to develop a model of shear capacity to provide references for the design and selection of BFRP connectors in engineering applications.

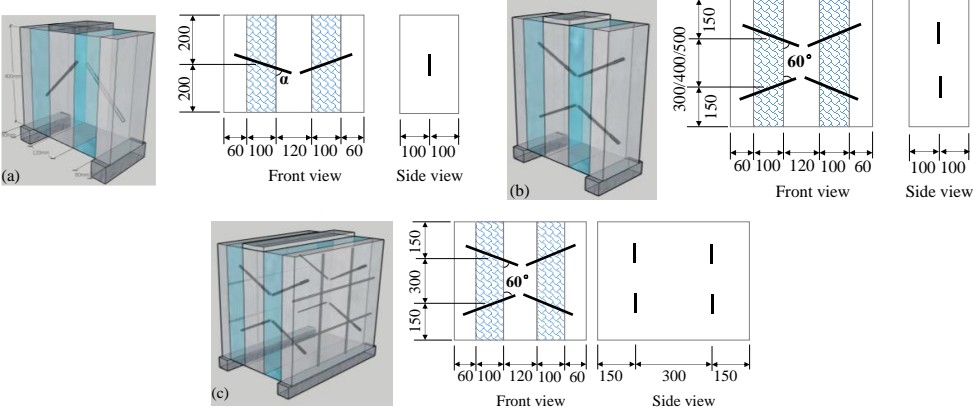

**Figure 1.** Schematic diagram of specimens (**a**) Double shear specimens; (**b**) ZS-1, ZS-2 or ZS-3; (**c**) ZS-4.

## 2. Experimental Investigation

### 2.1. Specimen Design

A total of 58 specimens were introduced in this paper with 54 double shear specimens and 4 combined double shear specimens. The 54 specimens were divided into 18 groups, and each group had three duplicates. The specimens were designed with the height of 400 mm, width of 200 mm and thickness of 200 mm. Two concrete wythes with thickness of 60 mm and 120 mm were separated by a layer of insulation. All components were assembled by BFRP connectors, as shown in Figure 1a. The parameters of connectors' diameters (12 mm, 10 mm and 8 mm), installation angles (90°, ±75°, ±60° and ±45°), stress modes (bending-shear, tension-bending-shear and compression-bending-shear) and the thicknesses of the insulation layer (100 mm, 150 mm, 200 mm and 250 mm) were studied.

Four groups of combined double shear specimens were introduced to study the bearing capacity of the combined connectors. Specimens of ZS-1, ZS-2 and ZS-3 with connectors' spacing of 300 mm, 400 mm and 500 mm were shown in Figure 1b. Four pairs of connectors in ZS-4, as shown in Figure 1c, exhibit a spatial combination with both vertical and transverse spacing of 300 mm.

Detailed test parameters of each group are listed in Table 1, where B/T/C in B/T/C-A1-A2-A3-A4 means stress modes of bending-shear (B) or tension-bending-shear (T) or compression-bending-shear (C), A1 means the diameter of connector, A2 means the installation angle, A3 means thickness of insulation layer and n means the order of duplicates.

**Table 1.** Parameter of specimens.

| Groups. | Specimen No. | d/mm | α | B/mm | Mechanical Modes |
|---|---|---|---|---|---|
| 1 | B-12-90-100-n | 12 | 90° | 100 | |
| 2 | B-10-90-100-n | 10 | 90° | 100 | Bending-shear |
| 3 | B-8-90-100-n | 8 | 90° | 100 | |
| 4 | T-10-75-100-n | 10 | −75° | 100 | |
| 5 | T-10-60-100-n | 10 | −60° | 100 | Tension-bending-shear |
| 6 | T-10-45-100-n | 10 | −45° | 100 | |
| 7 | C-10-75-100-n | 10 | 75° | 100 | |
| 8 | C-10-60-100-n | 10 | 60° | 100 | Compression-bending-shear |
| 9 | C-10-45-100-n | 10 | 45° | 100 | |

**Table 1.** *Cont.*

| Groups. | Specimen No. | d/mm | α | B/mm | Mechanical Modes |
|---|---|---|---|---|---|
| 10 | B-10-90-150-n | 10 | 90° | 150 | |
| 11 | B-10-90-200-n | 10 | 90° | 200 | Bending-shear |
| 12 | B-10-90-250-n | 10 | 90° | 250 | |
| 13 | T-10-60-150-n | 10 | −60° | 150 | |
| 14 | T-10-60-200-n | 10 | −60° | 200 | Tension-bending-shear |
| 15 | T-10-60-250-n | 10 | −60° | 250 | |
| 16 | C-10-60-150-n | 10 | 60° | 150 | |
| 17 | C-10-60-200-n | 10 | 60° | 200 | Compression-bending-shear |
| 18 | C-10-60-250-n | 10 | 60° | 250 | |
| 19 | ZS-1 | 10 | ±60° | 100 | Combination |
| 20 | ZS-2 | 10 | ±60° | 100 | Combination |
| 21 | ZS-3 | 10 | ±60° | 100 | Combination |
| 22 | ZS-4 | 10 | ±60° | 100 | Combination |

Note: d was the connectors' diameter, mm; α was the installation angle of the connector. B was the thickness of insulation layer, mm.

## 2.2. Materials Properties

The insulated PCSP system was composed of the FRP connector, concrete wythes and insulation layer. Previous experimental studies have shown that the bond performance of FRP bars was affected by concrete strength, bond length, external surface shape and other parameters [24–26]. Moreover, BFRP bars with rib spacing of 1 times the diameter, rib height of 6% diameter and rib width of 25% diameter have 29.5% higher bond strength than that of steel reinforcements [27]. Therefore, the deep ribbed BFRP bars (Figure 2) with thermal conductivity of 0.6 W/(m·K) were introduced as a connector in this study. The BFRP connectors were produced by Jiangsu GMV New Material T&D Co. Ltd. Nanjing, China, and their mechanical properties were shown in Table 2.

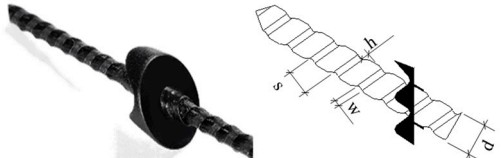

**Figure 2.** Typical BFRP connector.

**Table 2.** Mechanical properties of BFRP connector.

| Diameter/mm | s/mm | h/mm | w/mm | Tensile Strength/MPa | Elastic Modulus/GPa | Shear Strength/MPa | Inter-Laminar Shear Strength/MPa |
|---|---|---|---|---|---|---|---|
| 12 | 12 | 0.72 | 3.0 | 1032 (±27.5) | 52.3 (±1.40) | 204 (±3.88) | 37.8 (±0.81) |
| 10 | 10 | 0.60 | 2.5 | 1225 (±22.8) | 52.9 (±1.34) | 183 (±3.65) | 41.8 (±1.66) |
| 8 | 8 | 0.48 | 2.0 | 1179 (±19.9) | 53.4 (±0.66 | 213 (±8.77) | 48.3 (±2.07) |

Note: s means rib spacing; h means rib height; w means rib width. Numerical values in brackets are corresponding standard deviations.

Commercial concrete with a concrete grade of C25 was used. The compression test was carried out on three cubes after 28 days of curing under the same conditions with test specimens. The measured compressive strengths were 30.2 MPa, 28.4 MPa and 28.7 MPa, respectively. As the cubes were cast with a size of 100 mm × 100 mm × 100 mm, the compressive strength was multiplied by the correction factor of 0.95 according to "Standard for test method of mechanical properties on ordinary concrete" [28]. The measured average compressive strength of concrete was 27.7 MPa. The extruded polystyrene (XPS) with density of 35 kg/m$^3$, compressive strength of 250 kPa and thermal conductivity of 0.028 W/(m·K) was adopted as the insulation layer.

### 2.3. Fabrication and Casting

The manufacturing process could be summarized as four main steps, including preparation of connectors, mold manufacturing, installation and concrete casting. The BFRP connectors were inserted into the insulation layer, as shown in Figure 3a, and the vertical projection length at both sides was set to be 50 mm. Then, the insulation layer was installed into the formwork with positioning plates (Figure 3b). Finally, the concrete was cast and cured for 28 days with humidity (Figure 3c).

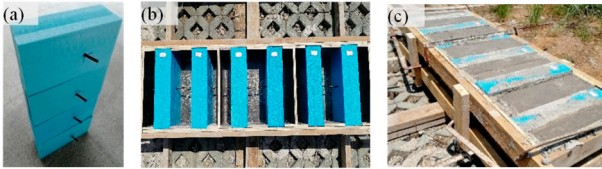

**Figure 3.** Manufacturing process: (**a**) Installation of connector; (**b**) Fix of insulation layer; (**c**) Concrete casting and curing.

### 2.4. Test Procedure and Loading Scheme

The test device is shown in Figure 4, where the outer two concrete wythes are supported on two steel blocks. B was the thickness of the insulation layer, and it was set to be 100 mm, 150 mm, 200 mm or 250 mm. The test was carried out on the actuator with a maximum load of 500 kN and accuracy of 0.5%. Stroke control loading mode was adopted with a speed of 2 mm/min [17]. The load was recorded by the sensor of the testing machine. The relative slip between the inner and outer wythes was measured by the displacement meter D1. During the loading process, the inclination of the outer wythe of the concrete was measured by the displacement meters D2 and D3. All the displacements were recorded by a data acquisition instrument.

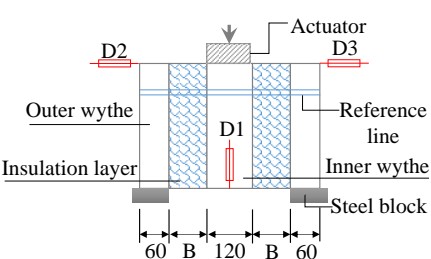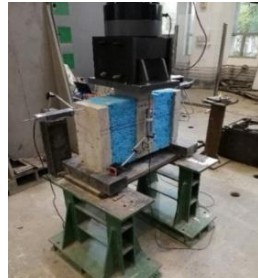

**Figure 4.** Loading scheme.

## 3. Results and Discussions

### 3.1. Failure Modes

The load capacity and failure modes are greatly affected by the connectors' diameter, installation angle, stress mode and thickness of the insulation layer. For direct observation, the insulation layer in all test specimens was removed, and the typical failure modes were shown in Figure 5.

For the bending-shear specimens, obvious relative slippages between the inner and outer wythes are observed, as shown in Figure 5(a), and then the specimens fail with inter-laminar shear of BFRP connectors, finally (Figure 5b). For tension-bending-shear specimens, the BFRP connectors also experience inter-laminar shear failure (Figure 5b) before the load reaches the peak. Finally, the concrete wythe fractures suddenly (Figure 5c) with the increasing of load.

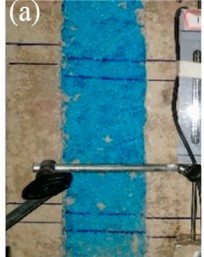 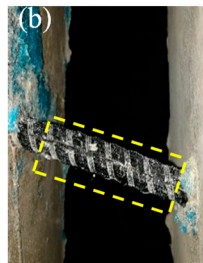 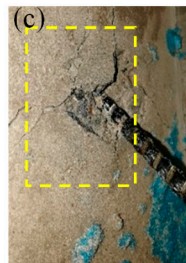 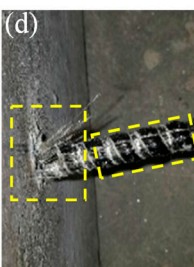 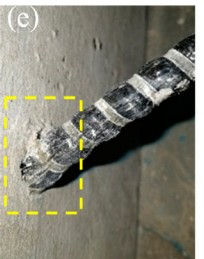 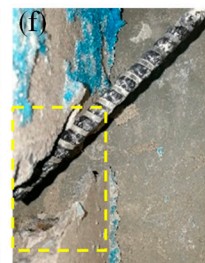

**Figure 5.** Failure modes (**a**) slippage; (**b**) inter-laminar failure; (**c**) concrete fracture; (**d**) inter-laminar failure and fiber fracture; (**e**) crack of connector; (**f**) concrete split.

Compared to tension-bending-shear specimens, the installation angle of BFRP connectors in compression-bending-shear specimens has great influence on failure modes. The BFRP connector with an installation angle of 75° experiences inter-laminar shear failure first, and then the top fiber bundle of BFRP connector fractures (Figure 5d). In addition, the BFRP connector with an installation angle of 60° cracks at its end (Figure 5e), while that of 45° keeps intact with a concrete split at its anchorage (Figure 5f). As concluded, the failure modes of BFRP connectors change from inter-laminar shear failure to the fracture failure and then to the anchorage failure with the decrease in the installation angle. This is attributed to the transformation of stress modes caused by different installation angles.

*3.2. Load-Displacement Curves*

Six typical groups of load-displacement curves were selected to study parameters of stress mode, span ratio and installation angle. As shown in Figure 6, the load-displacement curves of three duplicate specimens in each group keeps relatively consistent except for the compression-bending-shear specimens.

The load-displacement curves of the bending-shear specimens (Figure 6a,b) show obvious three-stage characteristics. The curves exhibit a good linear relationship in the first stage, where the insulation layer, the connector and outer concrete wythe work together. The curves' slopes decrease significantly when displacement reaches 3.4 mm for B-10-90-100, and 3.2 mm for B-10-90-150 specimens. This is attributed to the withdrawal of the insulation layer after the interface de-bonding between the insulation layer and outer concrete wythe, and the load was carried out by BFRP connectors. Then, the specimen enters the second stage where the stiffness decreases while maintaining a certain degree of linearity. At the third stage, the curves keep rising slowly until inter-laminar shear failure of the BFRP connector was observed.

Two-stage characteristics were evidenced in load-displacement curves for tension-bending-shear specimens, as shown in Figure 6c,d. Slipping tendency between the insulation layer and the wythes was observed at the beginning of the loading process. However, the specimen keeps excellent rigidity and the load increases linearly due to an "inverted splay" layout of BFRP connectors, where the outside fiber bundles in BFRP connectors were pulled, and then it makes full use of the advantage of the high tensile strength of the basalt fiber. At the second stage, the load displacement curve deflects sharply with decreased stiffness after the BFRP connector fails.

The BFRP connectors are subjected to compression and bending at the beginning of the loading process for the compression-bending-shear specimens, and the curves of C-10-45-100 and C-10-60-100 exhibit large, dispersed characteristics due to buckling of the connectors. The friction between the insulation layer and concrete wythes contributes to linearity of the load-displacement curve at the initial stage. However, the BFRP connector experiences sudden buckling when the load increases to a certain extent, resulting in a sharp decrease in load. Then, the load rises slowly after the specimen reaches a new stable state. The process was repeated until the test specimens fail with the fracture of the BFRP connector or the crushing of the outer concrete wythes.

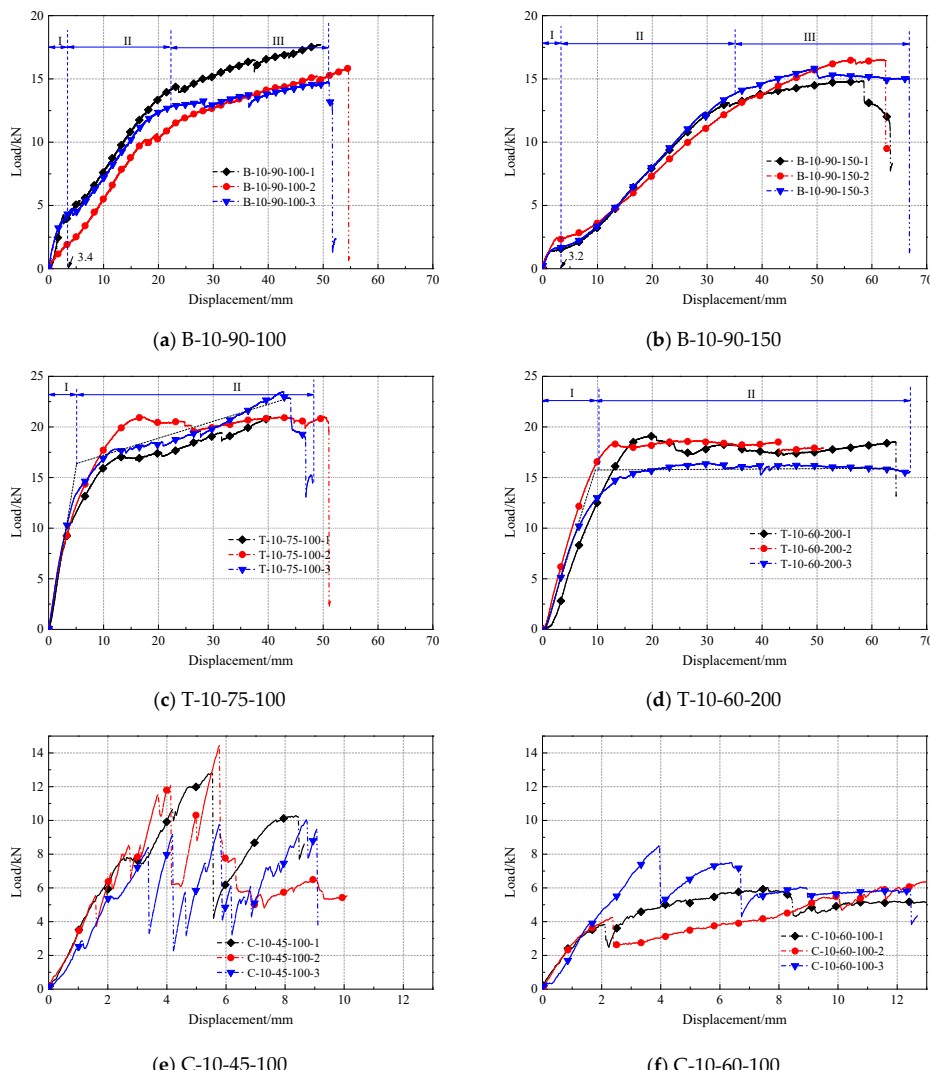

**Figure 6.** Load-displacement curves.

### 3.3. Parameter Analysis

As the double shear test specimens are designed symmetrically, half of the test value is adopted to characterize the load capacity of each single connector. The experimental results are listed in Table 3. According to ICC-ES AC320 [29] "Acceptance criteria for fiber-reinforced composite connectors anchored in concrete", the maximum displacement of inner and outer concrete wythes under external load shall not exceed 2.54 mm. Therefore, the shear bearing capacity $V_k$ in Table 3 was the load corresponding to the relative slippage of concrete wythes at 2.54 mm. $V_u$ was the maximum shear load, and $S_u$ was its corresponding slippage.

It can be seen that T-10-45-100 exhibited the highest shear capacity of 6.70 kN with the standard deviations of 0.071 kN, while that of B-10-90-250 was 0.35 kN ($\pm$0.177 kN). As indicated, parameters of diameter, the insulation layer's thickness, installation angle and layout spacing played important roles in load capacity. Therefore, detailed parameter discussions were listed below.

**Table 3.** Load capacity and corresponding displacement.

| Group | Specimen No. | $V_k$/kN | $V_u$/kN | $S_u$/mm |
|---|---|---|---|---|
| 1 | B-12-90-100 | 1.42 (±0.141) | 7.01 (±0.608)) | 16.1 (±0.735) |
| 2 | B-10-90-100 | 1.05 (±0.266) | 6.58 (±1.416) | 22.8 (±0.462) |
| 3 | B-8-90-100 | 0.77 (±0.113) | 5.09 (±0.283) | 20.6 (±1.061) |
| 4 | T-10-75-100 | 3.93 (±0.192) | 7.44 (±0.651) | 8.29 (±0.653) |
| 5 | T-10-60-100 | 4.30 (±0.020) | 7.85 (±0.896) | 6.11 (±0.642) |
| 6 | T-10-45-100 | 6.70 (±0.071) | 11.6 (±0.354) | 5.72 (±0.255) |
| 7 | C-10-75-100 | 1.79 (±0.375) | 1.94 (±0.332) | 5.08 (±1.266) |
| 8 | C-10-60-100 | 1.83 (±1.499) | 3.05 (±2.364) | 3.17 (±0.790) |
| 9 | C-10-45-100 | 3.89 (±1.308) | 3.91 (±1.113) | 3.18 (±0.718) |
| 10 | B-10-90-150 | 0.74 (±0.277) | 5.22 (±0.351) | 26.1 (±4.362) |
| 11 | B-10-90-200 | 0.57 (±0.293) | 3.87 (±1.894) | 28.0 (±9.551) |
| 12 | B-10-90-250 | 0.35 (±0.177) | 2.89 (±1.846) | 33.6 (±10.11) |
| 13 | T-10-60-150 | 3.31 (±0.552) | 7.71 (±0.802) | 7.06 (±0.460) |
| 14 | T-10-60-200 | 2.48 (±0.351) | 7.08 (±1.320) | 11.1 (±1.236) |
| 15 | T-10-60-250 | 2.05 (±0.101) | 6.89 (±0.907) | 11.6 (±0.513) |
| 16 | C-10-60-150 | 1.31 (±0.840) | 2.71 (±1.750) | 4.56 (±2.238) |
| 17 | C-10-60-200 | 0.95 (±0.380) | 2.03 (±0.972) | 8.13 (±3.955) |
| 18 | C-10-60-250 | 0.45 (±0.338) | 1.51 (±0.611) | 4.42 (±1.510) |
| 19 | ZS-1 | 6.34 | 11.6 | 6.25 |
| 20 | ZS-2 | 6.25 | 10.7 | 6.34 |
| 21 | ZS-3 | 6.20 | 10.1 | 6.40 |
| 22 | ZS-4 | 12.84 | 22.7 | 6.36 |

Note: Numerical values in brackets are corresponding standard deviations.

### 3.3.1. Span Ratio

The span ratio (R) was adopted to comprehensively evaluate the influence of the connector's diameter and the thickness of the insulation layer on the shear capacity. It was defined by the ratio of the insulation layer's thickness (B) to the connector's diameter (d). Span ratios with installation of 90° and 60° were discussed as follows:

Connectors with Installation Angles of 90°

The failure modes and stiffness of specimen with different span ratios are listed in Table 4. Inter-laminar shear failure was observed when the span ratio is less than 12.5, and BFRP connectors experienced a combination failure of both inter-laminar and connector fracture at the span ratio of 12.5 for B-10-90-100. The specimen of B-8-90-100 with a span ratio of 15 failed with a connector fracture, while the connectors with a span ratio larger than 20 kept intact due to the stroke limit of the machine. Their typical failure modes are shown in Figure 7. As concluded, the transformation of failure modes occurs at the span ratio of 12.5.

**Table 4.** Failure modes and stiffness of specimen with different span ratios.

| Groups | Specimen No. | B/mm | d/mm | R = B/d | Failure Modes | Stiffness k/(kN/mm) |
|---|---|---|---|---|---|---|
| 1 | B-12-90-100 | 100 | 12 | 8.33 | Inter-laminar shear failure | 0.559 (±0.028) |
| 2 | B-10-90-100 | 100 | 10 | 10 | Inter-laminar shear failure | 0.413 (±0.052) |
| 3 | B-8-90-100 | 100 | 8 | 12.5 | Inter-laminar shear failure, connector fracture | 0.303 (±0.022) |
| 4 | B-10-90-150 | 150 | 10 | 15 | connector fracture | 0.291 (±0.054) |
| 5 | B-10-90-200 | 200 | 10 | 20 | displacement limit | 0.224 (±0.057) |
| 6 | B-10-90-250 | 250 | 10 | 25 | displacement limit | 0.138 (±0.034) |

Note: The secant stiffness $k_1$ was defined as $k = V_k/2.54$, where 2.54 was the relative slippage between inner and outer concrete wythes, and $V_k$ was the corresponding load. Numerical values in brackets are corresponding standard deviations.

The shear capacity experiences non-linear degradation with span ratio, as shown in Figure 8, and it decreases 75.4% from a span ratio of 8.33 to 25. In addition, the structure stiffness decreases with the span ratio, as shown in Figure 9. The curves exhibit an obvious turning point at a span ratio of 12.5, and the stiffness at a span ratio of 15 (0.291 kN/mm) decreases by 4.0%, while that at a span ratio of 20 (0.224 kN/mm) decreases by 26.0%. In order to give full play to the material strength of BFRP connectors and obtain relatively

high stiffness, the span ratio for BFRP connectors with an installation angle of 90° was suggested to not be larger than 15.

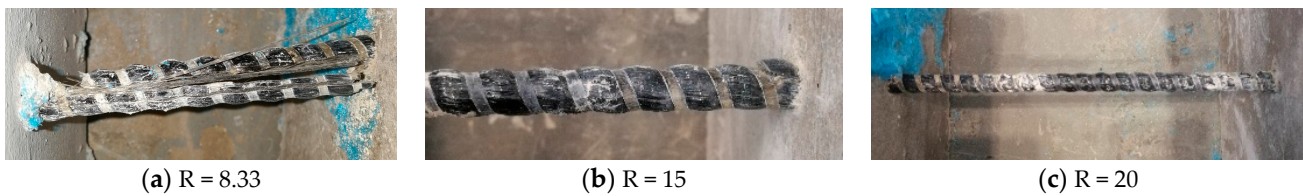

(**a**) R = 8.33        (**b**) R = 15        (**c**) R = 20

**Figure 7.** Failure modes of connectors with installation angle of 90°.

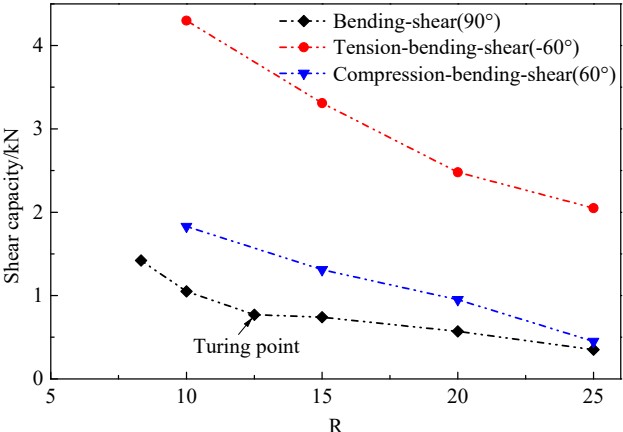

**Figure 8.** Degradation curves of shear capacity.

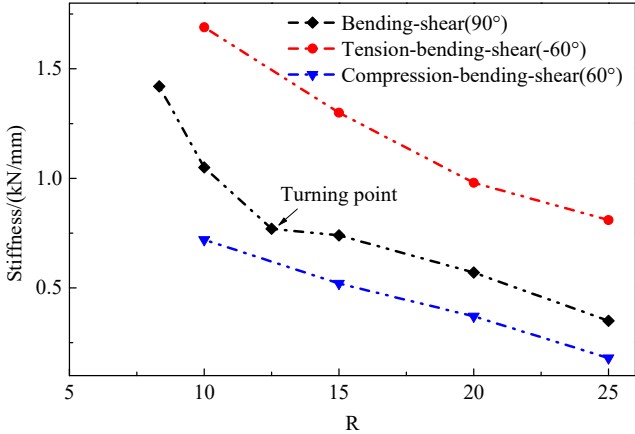

**Figure 9.** Degradation curves of stiffness.

Connectors with Installation Angles of ±60°

The failure modes and stiffness of specimen with different span ratios are listed in Table 5. As indicated, the stress modes could be classified into tension-bending-shear and compression-bending-shear for specimens with installation angles of ±60°. For tension-bending-shear specimens, the connectors experience inter-laminar shear failure when the span ratio is 10. The failure mode transforms from inter-laminar shear failure to connector fracture at the span ratio of 15. The connectors with a span ratio larger than 20 remain intact due to the stroke limit of the machine. However, for compression-bending-shear specimens, the specimens experience failure of connector fracture independent of span ratios.

**Table 5.** Failure modes and stiffness of specimen with different span ratios.

| Groups | Specimen No. | B/mm | d/mm | R=B/d | Failure Modes | Stiffness k (kN/mm) |
|---|---|---|---|---|---|---|
| 1 | T-10-60-100 | 100 | 10 | 10 | inter-laminar shear failure | 1.69 (±0.004) |
| 2 | T-10-60-150 | 150 | 10 | 15 | connector fracture | 1.30 (±0.109) |
| 3 | T-10-60-200 | 200 | 10 | 20 | displacement limit | 0.98 (±0.069) |
| 4 | T-10-60-250 | 250 | 10 | 25 | displacement limit | 0.81 (±0.020) |
| 5 | C-10-60-100 | 100 | 10 | 10 | connector fracture | 0.72 (±0.295) |
| 6 | C-10-60-150 | 150 | 10 | 15 | connector fracture | 0.52 (±0.165) |
| 7 | C-10-60-200 | 200 | 10 | 20 | connector fracture | 0.37 (±0.075) |
| 8 | C-10-60-250 | 250 | 10 | 25 | connector fracture | 0.18 (±0.066) |

Note: Numerical values in brackets are corresponding standard deviations.

As shown in Figures 8 and 9, both the shear capacity and stiffness experience approximate linear decrease with the span ratio. The maximum shear capacity for the compression-bending-shear specimen in C-10-60-100 was 1.83 kN, and it was smaller than the minimum shear capacity for tension-bending-shear specimen of T-10-60-250 (2.05 kN). Moreover, when the span ratio increases from 10 to 25, the structural stiffness decreases by 52% for tension-bending-shear specimens and decreases by 75% for the compression-bending-shear specimen.

As concluded, the span ratio for BFRP connectors was suggested to not be larger than 15 to obtain reasonable strength and stiffness of BFRP connectors. In fact, the displacement of inner and outer concrete wythes is proportional to the third power of span according to the conventional beam deflection equation. Thus, the material will not be fully utilized with a large span ratio.

### 3.3.2. Installation Angle

Figure 10 shows the comparison of load-displacement curves of tension-bending-shear and compression-bending-shear specimens with different installation angles. The diameter of BFRP connectors was 10 mm and the thickness of the insulation layer was 100 mm in the selected specimens. For installation angles of 75°, 60° and 45°, the shear capacity of the specimen under the compression-bending-shear mode was 26%, 39% and 34% that of tension-bending-shear mode. This is attributed to transformation of the stress mode caused by different installation angles. A previous study showed that the heterogeneous BFRP possessed different tensile and compressive properties, whose compressive strength was only 40% of the tensile strength [30].

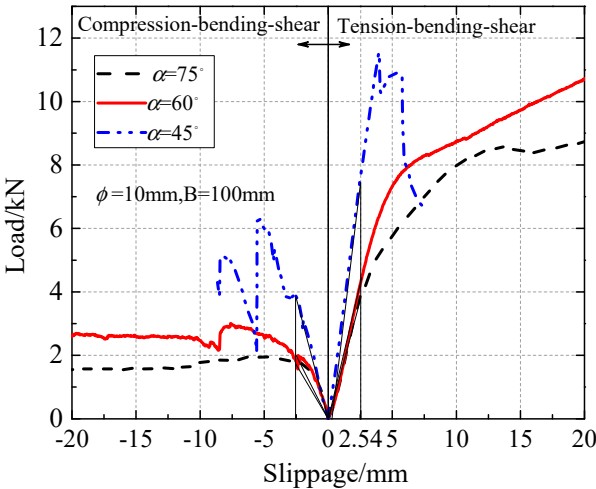

**Figure 10.** Comparison of load-displacement curves.

The stiffness comparison of specimens under different installation angles was shown in Figure 11. Generally, the stiffness increases with the decrease in the installation angle. Moreover, the stiffness of a specimen reinforced with connectors with an installation angle of −45° (2.64 kN/mm) and 45° (1.53 kN/mm) was 6.4 times and 3.7 times that of 90° (0.41 kN/mm), respectively. However, the connectors with installation angles of ±45°

experience brittle failure at small displacement (Figure 6c,f). Due to its small ductility and inadequate safety margin, the installation angles of ±45° should not be used for BFRP connectors. Therefore, the installation angles of 60° or 75° were recommended for suitable stiffness and shear bearing capacity.

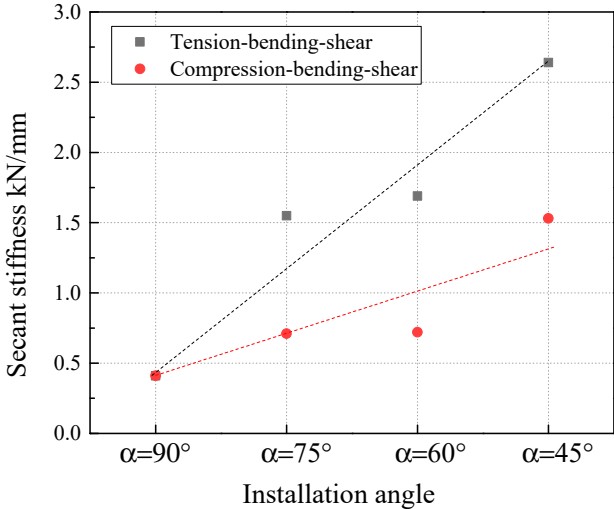

**Figure 11.** Comparison of stiffness.

The tensile property of the BFRP connector is better than the compression property; therefore, the utilization efficiency of the material was reduced during the transformation process from tension to compression, resulting in a reduction of ultimate bearing capacity.

### 3.3.3. Layout Spacing

The spacing of combined double shear specimens in ZS-1, ZS-2 and ZS-3 were 300 mm, 400 mm and 500 mm, respectively. The corresponding shear bearing capacity of ZS-1, ZS-2 and ZS-3 was 6.34 kN, 6.25 kN and 6.20 kN. The maximum difference of shear capacity between ZS-1 and ZS-3 was 2.2%. As indicated, the shear capacity decreases slightly with the increase in layout spacing of the connectors.

Moreover, the maximum shear load of ZS-1, ZS-2 and ZS-3 was 11.6 kN, 10.7 kN and 10.1 kN. The maximum shear load of ZS-1 improved 14.9% compared to that of ZS-3. The layout spacing of 300 mm improved the maximum shear load to an extent due to compact constraints of concretes between connectors. It can be concluded that the influence of layout spacing on shear capacity was relatively small with spacing ranging from 300 mm to 500 mm, while the maximum shear load improved 14.9% with the layout spacing of 300 mm.

### 3.3.4. Combined Action

BFRP connectors in combined test specimens of ZS-1, ZS-2 and ZS-3 were arranged symmetrically in the plane with an insulation angle of ±60°. The specimens were taken as a combination of T-10-60-100 and C-10-60-100. The errors of shear capacity between combined specimens and a single connector are listed in Table 6, where the $V_{kC}$ and $V_{kT}$ was the shear bearing capacity of C-10-60-100 and T-10-60-100 specimens. The values of $V_{kC}$ and $V_{kT}$ were 4.30 kN and 1.83 kN, according to Table 3. The $V_{uC}$ and $V_{uT}$ were the maximum shear load of C-10-60-100 and T-10-60-100 specimens, and their values were 7.85 kN and 3.05 kN, respectively. The maximum error of shear bearing capacity was 3.43% and that of the maximum shear load was −7.06%. As concluded, the shear capacity and the maximum shear load of a single connector could be used to design specimens with combined connectors.

Eight BFRP connectors in the combined test specimen of ZS-4 were arranged symmetrically in space, and it could be regarded as the superposition of two groups of ZS-1. Compared to the plane combination specimen of ZS-1, the shear bearing capacity of the spatial combination specimen of ZS-4 was approximately double. The errors of shear capacity

and ultimate bearing capacity were 1.26% and -2.29%, respectively. It can be concluded that the shear force of the plane or spatial combination specimen can be distributed according to that of a single BFRP connector.

**Table 6.** Errors of shear bearing capacity between combined specimens and single connector.

| Specimen No. | $V_k$/kN | $V_{kC} + V_{kT}$/kN | Error/% | $V_u$/kN | $V_{uC} + V_{uT}$/kN | Error/% |
|---|---|---|---|---|---|---|
| ZS-1 | 6.34 | 6.13 | 3.43 | 11.59 | 10.90 | 6.33 |
| ZS-2 | 6.25 | 6.13 | 1.96 | 10.67 | 10.90 | −2.11 |
| ZS-3 | 6.20 | 6.13 | 1.14 | 10.13 | 10.90 | −7.06 |
| ZS-4 | 12.84 | 12.68 | 1.26 | 22.65 | 23.18 | −2.29 |

*3.4. Prediction of Shear Capacity*

In order to facilitate the design and selection of BFRP connectors in practical implementation, the experimental results were analyzed by Isight. Quartic polynomial response surface model and cross-validation error analysis method were adopted. Therefore, the shear capacity $V_k$ corresponding to the relative slippage of concrete wythes at 2.54 mm was expressed as shown in Equation (1); and the maximum shear force $V_u$ was expressed as shown in Equation (2). The models of shear capacity and maximum shear force exhibited a highly identical degree of regression, and their correlation coefficients ($R^2$) were 0.982 and 0.946, respectively:

$$
\begin{aligned}
V_k = \ & 8 \times 10^{-5} R^4 - 43.7 \sin{(\alpha)}^4 + 1.88 \sin{(\alpha)}^3 - 0.0055 R^3 + 0.057 R \cdot \sin(\alpha) \\
& + 0.141 R^2 + 72.8 \sin{(\alpha)}^2 - 1.7 R - 3.43 \sin(\alpha) - 19.4
\end{aligned}
\tag{1}
$$

$$
\begin{aligned}
V_u = \ & 3.5 \times 10^{-4} R^4 + 284.1 \sin{(\alpha)}^4 - 0.0249 R^3 - 2.364 \sin{(\alpha)}^3 \\
& - 0.095 R \cdot \sin(\alpha) + 0.634 R^2 - 484.2 \sin{(\alpha)}^2 + 236.7
\end{aligned}
\tag{2}
$$

where $R$ was the span ratio and $\alpha$ was the installation angle.

The errors between the proposed model and experimental results are listed in Table 7. Specimens exhibit excellent accuracy except for those with an insulation layer thickness of 250 mm, where the errors of $V_k$ and $V_u$ were 130.5% and −103.4% at an installation angle of 60° and 46.4% and −28.7% at an installation angle of 90°. Their large errors were attributed to the failure modes of the stroke limit, which could not obtain the real material properties of connectors. Therefore, the proposed models were suggested to be applied for PCSP systems with an insulation layer thickness within 200 mm. Further studies on PCSP systems with an insulation layer thickness more than 200 mm need to be explored.

**Table 7.** Parameters of shear capacity model.

| No. | d/mm | B/mm | R | α/° | Experimental | | Proposed Model | | Error of $V_k$/% | Error of $V_u$/% |
|---|---|---|---|---|---|---|---|---|---|---|
| | | | | | $V_k$/kN | $V_u$/kN | $V_k$/kN | $V_u$/kN | | |
| 1 | 12 | 100 | 8.33 | 90 | 1.42 | 7.01 | 1.45 | 7.26 | 2.3 | 0.7 |
| 2 | 10 | 100 | 10 | 90 | 1.05 | 6.58 | 1.12 | 6.19 | 6.7 | −8.9 |
| 3 | 8 | 100 | 12.5 | 90 | 0.77 | 5.09 | 0.85 | 5.53 | 11.0 | 4.7 |
| 4 | 10 | 100 | 10 | −75 | 3.93 | 7.44 | 3.95 | 7.55 | 0.6 | −1.0 |
| 5 | 10 | 100 | 10 | −60 | 4.3 | 7.85 | 4.27 | 8.00 | −0.7 | 0.1 |
| 6 | 10 | 100 | 10 | 75 | 1.79 | 1.94 | 1.82 | 2.07 | 1.5 | −3.0 |
| 7 | 10 | 100 | 10 | 60 | 1.83 | 3.05 | 1.76 | 3.84 | −3.9 | 21.0 |
| 8 | 10 | 150 | 15 | −60 | 3.31 | 7.71 | 3.34 | 7.94 | 0.8 | 1.1 |
| 9 | 10 | 200 | 20 | −60 | 2.48 | 7.08 | 2.58 | 7.62 | 4.0 | 5.5 |
| 10 | 10 | 250 | 25 | −60 | 2.05 | 6.89 | 2.07 | 6.73 | 0.9 | −4.5 |
| 11 | 10 | 150 | 15 | 60 | 1.31 | 2.71 | 1.32 | 2.96 | 0.7 | 3.5 |
| 12 | 10 | 200 | 20 | 60 | 0.95 | 2.03 | 1.05 | 1.81 | 10.8 | −18.3 |
| 13 | 10 | 250 | 25 | 60 | 0.45 | 1.51 | 1.04 | 0.10 | 130.5 | −103.4 |
| 14 | 10 | 150 | 15 | 90 | 0.74 | 5.22 | 0.72 | 5.25 | −3.0 | −3.3 |
| 15 | 10 | 200 | 20 | 90 | 0.57 | 3.87 | 0.49 | 4.03 | −14.0 | −0.9 |
| 16 | 10 | 250 | 25 | 90 | 0.35 | 2.89 | 0.51 | 2.26 | 46.4 | −28.7 |

## 4. Conclusions

PCSP systems have been the preferred approach due to accelerated building construction, reduced construction costs and enhanced performance. Moreover, the adoption of nonmetallic connectors such as the BFRP connector can satisfy the structural insulation requirements due to its low thermal conductivity. The parameter study on BFRP connectors in this paper could provide an alternative option for insulated PCSP systems reinforced with FRP connectors.

Based on the double shear tests of insulated PCSP systems reinforced with BFRP connectors, the failure modes, load displacement curves and load capacity were obtained. The influences of BFRP connectors' span ratio, installation angle, layout spacing and combined action on the shear capacity were analyzed. The models for prediction of shear capacity and the maximum shear force were proposed. The following conclusions can be summarized:

(1) BFRP connectors experience typical failure modes changing from inter-laminar shear failure to the connector fracture and then to the concrete split with the decrease in the installation angle. This is attributed to the transformation of stress modes caused by different installation angles.

(2) Stress modes have a great effect on discreteness of the load-displacement curves. Those of bending-shear and tension-bending-shear keep consistent, while that of the compression-bending-shear exhibits large, dispersed characteristics due to buckling of connectors.

(3) The BFRP connector's span ratio of not larger than 15 and installation angles of 60° or 75° were recommended for suitable stiffness and shear bearing capacity. The influence of layout spacing between 300 mm and 500 mm on shear capacity was relatively small, where the maximum error was 2.2%. The material performance will be fully utilized with suggested parameters for practical implementation.

(4) For the design of BFRP connectors with a plane or spatial combination, the shear force can be calculated by linear superposition of a single BFRP connector's shear force. It facilitates the design of an insulated PCSP system with various combinations.

(5) The prediction models for shear capacity and the maximum shear force exhibit good accuracy with correlation coefficients ($R^2$) of 0.982 and 0.946. It could provide a basis for the design and selection of BFRP connectors, especially for an insulated PCSP system with thick insulation layers within 200 mm.

This paper provides a preliminary study on the shear behavior of insulated precast concrete sandwich panels reinforced with BFRP connectors. Parameters including the span ratio, installation angles and layout spacing have been suggested for BFRP connectors in future engineering applications. The proposed models for shear capacity and the maximum shear force could provide a favorable design option of an insulated PCSP system by conforming to the required design criteria.

**Author Contributions:** Conceptualization, X.L. and X.W.; methodology, X.W.; software, X.L.; validation, X.L., X.W. and Z.W.; formal analysis, T.Y.; investigation, T.Y.; resources, T.Y.; data curation, T.Y.; writing—original draft preparation, X.L. and X.W.; writing—review and editing, X.L. and X.W.; visualization, X.L.; supervision, X.W. and Z.W.; project administration, X.W. and Z.W.; funding acquisition, X.W. and Z.W. All authors have read and agreed to the published version of the manuscript.

**Funding:** This research was funded by the National Key Research and Development Program of China (2017YFC0703000), the Fundamental Research Funds for the Central Universities (2242022k30031 and 2242022k30033).

**Institutional Review Board Statement:** Not applicable.

**Informed Consent Statement:** Not applicable.

**Data Availability Statement:** The data presented in this study are available in article.

**Conflicts of Interest:** The authors declare no conflict of interest.

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
