# Peer review of "The Shear Behavior of Insulated Precast Concrete Sandwich Panels Reinforced with BFRP"

_buildings, doi:10.3390/buildings12091326_

Round 1
Reviewer 1 Report
The work is well written and structured. The results are very interesting and properly presented, and the conclusions are interesting within the scope of this journal. I miss a comparison, even theoretical, with the same samples but with steel rebars instead of the used in the analysis. With this result, it would be easy to evaluate the performance of the proposed configuration. The reader could benefit from that, since will have a clear picture of the benefits and drawbacks of the proposed methodology, engaging its potential transfer towards a industrial implementation.
Reviewer 2 Report
The conducted work “The shear behavior of insulated Precast Concrete Sandwich Panels reinforced with BFRP” is good. However, following comments should be addressed to further improve paper:
A. GENERAL COMMENTS FOR PAPER ON OVERALL BASIS
1. Explicitly mention the novelty and research significance of current work in last paragraph of introduction section with emphasis on scientific soundness. Also, add recent relevant literature review from 2021 and 2022 papers in introduction section as there is no paper from 2021 and 2022.
2. Heading 2.4 should be “Test Procedure and Loading Scheme”.
3. Results inn current form look like a lab report. Results should be further elaborated with scientific reasoning.
4. A separate brief section (explaining the relevance of this research for practical implementation) may be added before conclusion section.
5. Closing remarks should be added at the end of conclusion section keeping in mind all conclusive bullet points.
6. English Language should be improved throughout the manuscript.
B. SPECIFIC COMMENTS FOR IMPROVING FOCUSSED RESEARCH
1. Show standard deviation where-ever the average is being taken.
2. Why correction factor of 0.95 is being used?
3. Figure 6: the ranges for x and y axes should be same in all graphs shown in a to f so that comparison will be easy to understand.
